# MicroRNA-708 emerges as a potential candidate to target undruggable NRAS

Jia Meng Pang[1,2☯], Po-Chen Chien[2☯], Ming-Chien Kao[2], Pei-Yun Chiu[3], Pin-Xu Chen[2], Yu-Ling Hsu[1,2], Chengyang Liu[4], Xiaowei Liang[4], Kai-Ti Lin[1,2]*

1 Department of Medical Science, National Tsing Hua University, Hsinchu, Taiwan, 2 Institute of Biotechnology, National Tsing Hua University, Hsinchu, Taiwan, 3 Interdisciplinary Program of Life Science, National Tsing Hua University, Hsinchu, Taiwan, 4 College of Life Sciences, Wuhan University, Wuhan, Hubei, China

☯ These authors contributed equally to this work.
* ktlin@life.nthu.edu.tw

**Data Availability Statement:** All relevant data are within the paper and its Supporting information files.

**Funding:** This research was funded National Science and Technology Council (NSTC) (108-

## Abstract

*RAS*, the most frequently mutated oncogene that drives tumorigenesis by promoting cell proliferation, survival, and motility, has been perceived as undruggable for the past three decades. However, intense research in the past has mainly focused on *KRAS* mutations, and targeted therapy for *NRAS* mutations remains an unmet medical need. *NRAS* mutation is frequently observed in several cancer types, including melanoma (15–20%), leukemia (10%), and occasionally other cancer types. Here, we report using miRNA-708, which targets the distinct 3' untranslated region (3'UTR) of *NRAS*, to develop miRNA-based precision medicine to treat *NRAS* mutation-driven cancers. We first confirmed that *NRAS* is a direct target of miRNA-708. Overexpression of miRNA-708 successfully reduced NRAS protein levels in melanoma, leukemia, and lung cancer cell lines with *NRAS* mutations, resulting in suppressed cell proliferation, anchorage-independent growth, and promotion of reactive oxygen species-induced apoptosis. Consistent with the functional data, the activities of NRAS-downstream effectors, the PI3K-AKT-mTOR or RAF-MEK-ERK signaling pathway, were impaired in miR-708 overexpressing cells. On the other hand, cell proliferation was not disturbed by miRNA-708 in cell lines carrying wild-type *NRAS*. Collectively, our data unveil the therapeutic potential of using miRNA-708 in *NRAS* mutation-driven cancers through direct depletion of constitutively active NRAS and thus inhibition of its downstream effectors to decelerate cancer progression. Harnessing the beneficial effects of miR-708 may therefore offer a potential avenue for small RNA-mediated precision medicine in cancer treatment.

## Introduction

*RAS* genes, including *KRAS*, *HRAS*, and *NRAS*, encode a small GTPase that acts as a binary switch, toggling between the GDP-bound inactive state and GTP-bound active state, to communicate environmental growth factor cues and the downstream signaling network to control the cell cycle, differentiation, and survival [1]. Up to 30% of all human cancers carry oncogenic

2314-B-007-003-MY3 and 111-2320-B-007-005-MY3) and National Tsing Hua University (111Q2713E1, 112Q2511E1, and 112Q2521E1). The funders had no role in study design, data collection and analysis, decision to publish, or preparation of the manuscript.

mutations in one of these three *RAS* isoforms [2]. Codons 12, 13, and 61 are the hotspots for amino acid substitution mutations, trapping RAS protein in the GTP-bound active state [3]. The inhibition of GTP hydrolysis results in constitutively active RAS, leading to uncontrolled cell growth and malignant transformation of cells [4].

*RAS* is one of the earliest identified oncogenes [5] and the whole world has spurred intense resources in developing RAS inhibitors for over 40 years. However, no effective RAS inhibitor was approved in clinical practice until 2021. As a result, RAS was even perceived as an "undruggable" target for many years [6]. This traditional perception has been revolutionized with the pioneering research done by Shokat and colleagues, who paved the way for the design of some promising allele-specific covalent inhibitors (e.g. AMG510 and MRTX849) to target the $KRAS^{G12C}$ mutant [7, 8]. Unfortunately, this discovery is restricted only to the $KRAS^{G12C}$-mutant and cannot be translated into other *RAS* isoform-mediated oncogenesis.

Given that *KRAS* is predominantly mutated in cancers harboring *RAS* mutations, drug development targeting *NRAS* mutations lags far behind and remains an unmet medical need. *NRAS* is mutated in cutaneous melanoma and acute myeloid leukemia (AML) at 20% and 15% frequencies respectively. Some other cancers including lung cancer, colon cancer, and neuroblastoma are also reported to carry *NRAS* mutations at a lower frequency [2]. Melanoma cells with *NRAS* mutations have elevated mitotic activity, causing thicker lesions and a higher rate of metastasis as a consequence of persistent NRAS signaling [9]. Melanoma patients harboring this mutation present poor prognostic outcomes compared to patients harboring other mutations [10]. *NRAS* mutant-mediated aggressiveness is not only limited to melanoma but its correlation with the higher mortality rate in pediatric AML is also observed [11]. Nonetheless, direct inhibition of NRAS activity is extremely challenging, and no clinical trial has been conducted to directly target NRAS. The lack of deep hydrophobic pockets on the NRAS catalytic G domain, the high binding affinity to GTP, and identical G domains among different RAS isoforms all pose obstacles in developing efficacious NRAS-selective drugs [1]. Current management of *NRAS* mutant melanoma mainly relies on MEK inhibitors (MEKi) and immune checkpoint blockade (ICB) therapy [12]. However, third-generation MEKi, such as trametinib and binimetinib, only showed a modest response rate when used alone [13, 14]. MEKi in combination with CDK4/6 or PI3K/AKT inhibitors showed enhanced antitumor activity but also caused undesirable toxicities in clinical trials [15, 16]. Instead, ICB therapy shows promising results for advanced metastatic melanoma with *NRAS* mutation [17–20]. However, some patients who receive ICB therapy develop resistance [21] or recurrence after treatment [22]. Therefore, developing drugs selectively targeting *NRAS* mutants is urgently needed.

Since pan-RAS inhibitors may lead to substantial toxicity as the RAS signaling pathway accounts for many critical cellular events in noncancerous cells [23], isoform-specific inhibitors should be taken into consideration when drugging NRAS. Here, we offer a workable solution to circumvent off-target toxicities and deplete the NRAS protein effectively by using a microRNA-based approach. MicroRNA (miRNA) is a short noncoding RNA that can negatively regulate gene expression posttranscriptionally [24, 25]. It binds generally to the 3'-untranslated regions (3'UTR) of target mRNA and induces translational repression, degradation, or cleavage [26, 27]. To date, dysregulated miRNAs are associated with tumor development, progression, metastasis, and response to therapy, suggesting their potential use as diagnostic, prognostic, and predictive biomarkers [28, 29]. Meanwhile, in light of the first siRNA drug approved by the FDA in 2018 [30], the number of miRNA drugs entering or getting closer to clinical trials has increased [31]. Harnessing the power of miRNA may help us to develop a therapeutic strategy to suppress those "undruggable proteins" through epigenetic inhibition [32]. In the case of *RAS*, different isoforms share more than 90% homology in their protein sequences; hence, targeting sequences in the 3'untranslated region (3'UTR) of *RAS*

isoforms by miRNA can result in unique inhibition of each individual isoform of *RAS* and thus provide a great opportunity to specifically deplete NRAS protein expression.

MicroRNA-708 (miR-708), a tumor suppressor [33], is located within the first intron of the ODZ4 gene [34]. The discovery of miR-708 has drawn oncologists' attention to its role in suppressing cancer progression. Studies have reported that downregulation of miR-708 correlates with the occurrence, advanced stage, and poor survival in a variety of different malignancies while restoring miR-708 expression in different cancer cells results in decreased cell migration and invasion *in vitro*, as well as reduced tumor growth and metastasis *in vivo* [35–40]. Few targets of miR-708 have been proven to participate in cell motility, including Rap1B in ovarian cancer [35], KPNA4 in prostate cancer [39], and NNAT in breast cancer [40]. MiR-708 restoration in those cancer cells significantly inhibited those targeted genes and eventually attenuated distant metastasis. In addition to acting as an antimetastatic factor in cancer, miR-708 also exerts a profound antiproliferative effect by suppressing prosurvival proteins (e.g. survivin [36], and Bcl-2 [38]), DNA repair mechanisms (e.g. BMI1 [36], and PARP1 [38]), and proteins related to the cell cycle, such as cyclin D1 [38], to suppress tumor growth. However, the role of miR-708 in melanoma and acute myeloid leukemia (AML) has yet to be characterized, and whether the expression of *NRAS*, one of the most important oncogenes in these cancers, is targeted by miR-708 remains unknown.

In this study, we identified and confirmed that miR-708 directly targets *NRAS* to deplete its protein levels in different cancer cell lines driven by *NRAS* mutation, resulting in decreased signaling in the effector pathways, PI3K-Akt-mTOR or RAF-MEK-ERK, and subsequently alleviating cancer cell proliferation, anchorage-independent growth, motility, and resistance to apoptosis. On the other hand, cell proliferation was not suppressed by miR-708 in cancer cells carrying wild-type *NRAS* or normal lung epithelial cells. Overall, our data suggest that miR-708 can be used as a promising precision medicine for cancers driven by *NRAS* mutations in the near future.

## Material & methods

### Cell culture

Human melanoma cell line SK-MEL-2, human breast cancer cell line MDA-MB-231, and human lung cancer cell line A549 were maintained in DMEM (Invitrogen) while acute myeloid leukemia cell line THP-1, Non-small cell lung cancer cell line H1299, and normal lung epithelial cell line BEAS-2B were cultured in RPMI (Invitrogen). All cancer cell lines were obtained from Bioresource Collection and Research Center (BCRC), Taiwan. BEAS-2B was a gift from Dr. I-Ching Wang of National Tsing Hua University (Taiwan). All culture media were supplemented with 10% fetal bovine serum (FBS; Biological Industries, Israel or Hyclone, USA) and 1% of penicillin-streptomycin (GIBCO), cultured at 37°c with 5% $CO_2$.

### Nucleotides and reagents

The precursor miR-708 was purchased from Applied Biosystem (Carlsbad, CA). The *NRAS*-specific siRNA and negative control siRNA (NC) were from MDBio, Inc (Qingdao, China). Detailed sequences of primers and small RNAs are listed in Table S1 in S1 File.

### Transfection and transduction

For transfection of adherent cells, 20 or 50nM of miR-708 precursor, *NRAS* siRNA, or NC siRNA were delivered using TransIT-X2 (Mirus Bio) or Lipofectamine® RNAiMax (Invitrogen) in Opti-MEM (Gibco) for 5h, then replaced with normal culture medium. For

transfection of suspension cells, THP-1 was electroporated using Neon® Transfection System at the condition recommended (1600V/10ms/3 pulses), and $10^6$ cells were transfected with 200nM miRNA or siRNA respectively in Buffer T provided. Transfected cells were harvested 48–72 h post-transfection for subsequent analysis.

## Quantitative real-time PCR for genes and miRNAs

RNA was extracted from cells using TRIzol (Invitrogen) according to the manufacturer's protocol. First-strand cDNA was synthesized by ReverTraAce (Toyobo, Japan) using oligo-dT or specific microRNA RT primers designed based on a previous study [41] from 1ug of total RNA. Real-time RT-PCR was performed on a qTower 3 real-time PCR Thermal Cycler (Analytik Jena). The KAPA SYBR FAST Universal qPCR Kit (KAPA Biosystems, MA) was used for *NRAS* detection. The KAPA PROBE FAST universal qPCR Kit together with Universal Probe Library no. 21 (Roche Applied Science) was used for miR-708 detection. The mRNA levels were normalized with actin, while miRNA levels were normalized with U6. All the primer sequences used in this study are provided in Table S1 in S1 File. Data are means ± SD from 3 independent experiments and presented as fold changes relative to the control.

## Western Blot analysis

Cells were lysed in 10mM Tris buffer, pH 7.4, containing 0.15 M NaCl, 1% Triton X-100, 100mM EDTA, protease inhibitor (Roche), and phosphatase inhibitor (Cyrusbio) mixture. The cell lysates were resolved in 10% SDS-Polyacrylamide gel, transferred onto PVDF membrane, and probed with antibodies. Antibodies against NRAS (sc-519) and β-actin (sc-47778) were purchased from Santa Cruz Biotechnology, while antibodies against Akt (pan) (4691S), phospho-Akt (Ser 473) (4060S), p44/42 MAPK (t-ERK 1/2) (9102S), and phospho-p44/42 MAPK (p-ERK 1/2) (Thr 202 and Tyr 204) (9101S) were purchased from Cell Signaling Technology. Protein expression levels were quantified with Image J (NIH, USA). The raw blot was quantified with uncalibrated OD. All bands within the same lane were selected with rectangular tools and plotted. The maximum background value of the membrane on the lane was then subtracted. The area of each peak was quantified by integration using Wand tools and regarded as protein expression levels. The expression of NRAS, p-AKT, and p-ERK1/2 were then normalized with Actin, Akt (pan), and ERK1/2 respectively. Data are means ± SEM from 3 independent experiments and presented as fold changes relative to the control.

## 3'UTR luciferase reporter assay

Sequences containing the predicted miR-708 binding site (157 bps) on *NRAS* mRNA 3'UTR region were amplified and cloned into the pGL3-Basic Vector (Promega). Primer sequences used for cloning are provided in Table S1 in S1 File. The mutant form of 3'UTR was generated using site-directed mutagenesis. In the 3'UTR luciferase reporter assay, SK-MEL-2 cells were transfected with 1ug of pGL3-NRAS 3'UTR (wild type or mutant), together with 0.1ug of GFP construct as the internal control. 24 h later, miR-708 was transfected into the cells. The cells were lysed with passive lysis buffer for the detection of luciferase signal with Dual-Luciferase Reporter System (Promega) or Neolite Reporter Gene Assay System (PerkinElmer, USA). The Luciferase and GFP activity were read by Synergy HTX (BioTek). Luciferase activity was normalized with GFP activity. Data are means ± SD from 3 independent experiments and presented as fold changes relative to the control.

## Cell proliferation assay

Cell proliferation was measured through the clonogenic assay. SK-MEL-2 cells ($3 \times 10^3$ cells), MDA-MB-231 cells ($3 \times 10^3$ cells), BEAS-2B ($0.5 \times 10^3$ cells), A549 ($1 \times 10^3$ cells), and H1299 cells ($1 \times 10^3$ cells) were seeded into 24-well plates in triplicate. After miR-708 transfection, cells were incubated for 7 or 9 days. Colonies were fixed and stained with 0.1% crystal violet for 30 min. Photos of triplicate wells were then scanned. Cell proliferation was quantified with Image J by a plugin called Colony Area developed by Manish Bagga et al [42]. The parameter used to illustrate the result is colony intensity percentage, defined by an equation:

$$\text{Colony intensity} \% = \frac{\sum pixel\ intensities\ in\ a\ region}{\sum maximum\ intensities\ possible\ in\ the\ same\ region} \times 100.$$

Data are means ± SD from 3 independent experiments and presented as fold changes relative to the mock control.

## Soft agar colony formation assay

To prepare for the experiment, the bottom agar layers were prepared by mixing 2x DMEM or RPMI (20% FBS and 2% penicillin/streptomycin supplemented) and 1.2% of sterile high-melting agarose (FocusBio) in a ratio of 1:1, then placed 350 μL of the mixture into a 24-well culture dishes and allowed it to be solidified in room temperature. $1 \times 10^3$ of SK-MEL-2 cells, H1299 cells, and THP-1 cells that transfected with miR-708 and siNRAS were harvested to perform the experiment in order to determine the anoikis and anchorage-independent growth of cells in soft agar. The harvested cells are resuspended in 100 μL of culture medium. Cells were then mixed with 100 μL of 2x DMEM or RPMI and 1.2% of sterile low-melting agarose (LONZA SeaPlaqueTM agarose) in a 1:1:1 manner. The plate was then incubated at 4°C for 5 min to solidify the middle layer of agar-containing cells. To prevent the culture from drying out, 350μL of culture medium was added to each well before transfer to the incubator. The cells were allowed to grow into colonies for one (THP-1) or two (SK-MEL-2, H1299) weeks. After that, cells were stained with iodonitrotetrazolium violet (INT) for 2 days. Images of cell colonies from 3 separate fields of 2 different vertical planes of each well were taken using a microscope with 40X and 100X magnification (IX51, Olympus). The number of colonies in each field of view was quantified using the cell counter plugin in ImageJ and summed up for each group. For area quantification, each colony was circled and the area (pixel$^2$) was analyzed by ImageJ as well. The mean colony area was obtained from the average of the total colony area. Data are means ± SD from 3 independent experiments and presented as fold changes relative to the mock control.

## Transwell cell migration and cell invasion assay

Cell migration was assayed in 8.0-mm Falcon Cell Culture Inserts (Corning). For the cell invasion assay, the BD biocoat Matrigel invasion chamber was applied (Corning) and performed as previously described [35]. Briefly, $2 \times 10^4$ to $1 \times 10^5$ of miR-708 or siNRAS-transfected SK-MEL-2 or H1299 cells were suspended in serum-free DMEM (500 μL), and placed in the upper transwell of 0.3 cm$^2$ in area. The bottom well was filled with 500 μL DMEM with FBS. After incubation for 16 h to 24 h, cells on the upper side of the inserts were removed by cotton swabs, and cells on the underside were fixed and stained with crystal violet. Photos of three (migration) to five (invasion) regions were taken from duplicated inserts, and the numbers of cells were counted using Image J. Data are means ± SD from 3 independent experiments and presented as fold changes relative to the mock control.

## Chemotaxis assay

Chemotaxis of THP-1 cells was examined by using 8.0-mm Falcon Cell Culture Inserts (Corning). $1 \times 10^5$ THP-1 cells transfected miR-708 were resuspended in serum-free RPMI 1640 (500 uL), and placed in the upper transwell of $0.3 \text{ cm}^2$ in area. The bottom well was filled with 500 μL DMEM with 10% FBS. After 24 h incubation, the inserts were removed and the migrated cells in the bottom well were stained with CellTracker™ Red CMTPX Dye (Invitrogen) for 20 minutes. Photos of five regions were taken from the duplicated bottom wells and the number of cells was counted automatically using ImageJ. Data are means ± SD from 3 independent experiments and presented as fold changes relative to the mock control.

## Apoptosis assay

Cells transfected with miR-708 or siNRAS were treated with 200 μM hydrogen peroxide ($H_2O_2$) (Showa Denko, Japan) to induce apoptosis for 24 hr after 48 hr transfection. Cells were then collected and double stained with 80 ng/mL Annexin V (BioLegend, 640920) and 5 μg/mL 7-AAD (BioLegend, 420403) in calcium-rich binding buffer (10mM pH 7.8 HEPES, 140mM NaCl, 2.5mM $CaCl_2$) for 15 minutes. The stained cells were then subjected to flow cytometry analysis (Beckman Coulter) and the raw data was analyzed by using CytExpert 2.0. Data are means ± SD from 3 independent experiments and presented as fold changes relative to the mock control.

## Survival correlation analysis

To investigate the correlation between miR-708 expression and overall survival of patients with different types of cancer, including Skin Cutaneous Melanoma Project (TCGA-SKCM), Acute Myeloid Leukemia Project (TCGA-LAML), and Lung Adenocarcinoma Project (TCGA-LUAD), we retrieved clinical data and miRNA expression quantification from The Cancer Genome Atlas (TCGA) database (http://cancergemome.nih.gov/). All data analyzed in this study were openly acquired from TCGA databases, therefore, ethics approval was not obtained and the need for informed consent was waived. Statistical analysis was carried out using R. The survival and survminer packages were used to model patient survival data. The "ggpubr" package was used to model scatter plot data. For survival data, we divided the patients into high and low-expression groups based on the median value of the miR-708 expression in the miRNA-sequencing file. The survival period of each patient was defined as the days to the last follow-up. Overall survival was demonstrated with the Kaplan-Meier method. Detailed clinical information is provided in S2 File.

## Statistical analyses

The statistical analyses were all assessed by GraphPad Prism using Student's t-tests to compare two means. ANOVA followed by Tukey's post-hoc test was used for the statistical analysis when more than two means were compared. The correlation significance between miR-708 expression and survival was tested by log-rank test.

# Results

## NRAS is the direct target of miR-708

To identify miRNAs that potentially deplete NRAS expression and subsequently suppress tumorigenesis driven by *NRAS* mutation, we applied TargetScan databases to retrieve miRNAs capable of binding the 3' UTR of *NRAS*. TargetScan is an online portal that is used to predict the potential alignments between miRNAs and mRNAs by using a computational quantitative

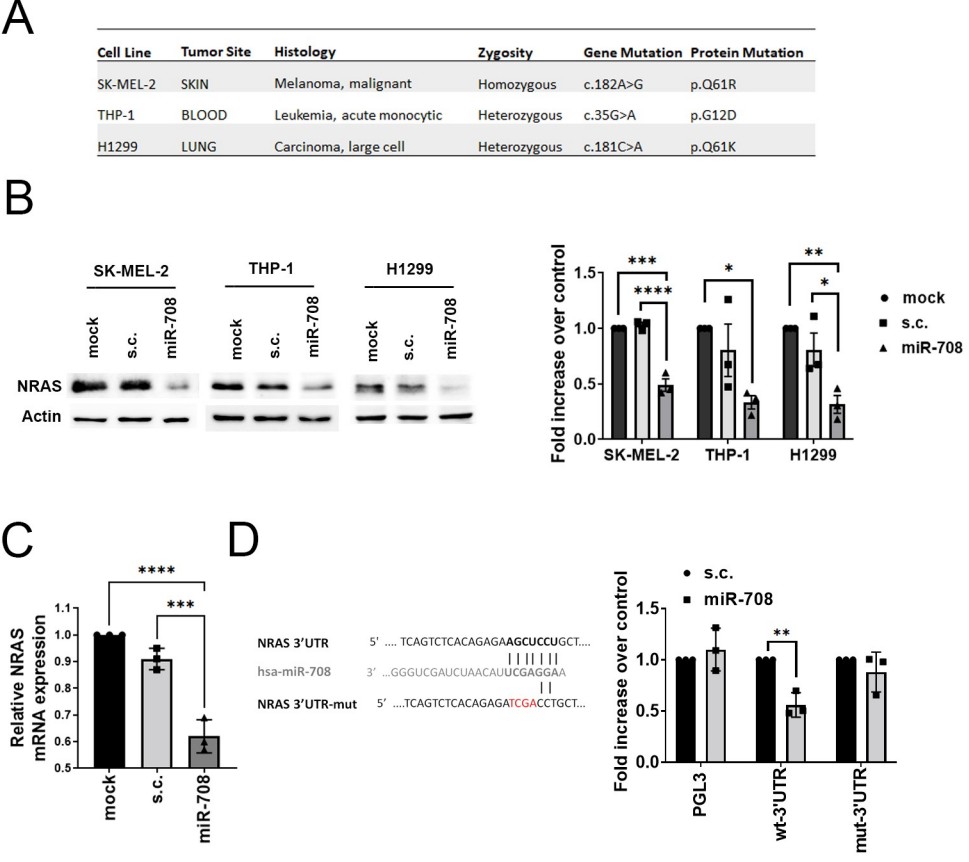

**Fig 1. NRAS is the direct target of miR-708.** (A) Cancer cell lines with NRAS mutation used in this study. (B) Left: Representative Western blot of NRAS protein expression in SK-MEL-2, THP-1, and H1299 cells transfected with mock, scrambled control (s.c.), and precursor miR-708 (pre-miR-708) for 48 h. Right: quantitative analysis of NRAS protein level, normalized with actin. Histogram represents normalized means ± SEM (n = 3 biological replicates). (C) Expression of NRAS mRNA in SK-MEL-2 cells transfected mock, s.c., and pre-miR-708. Data represent normalized means ± SD (n = 3 biological replicates). (D) Luciferase reporter assays showing luciferase activities in the pGL3 construct conjugated with NRAS 3'UTR (wild type or mutant form) by SK-MEL-2 transfected with pre-miR-708 or s.c. Left: the sequence of miR-708 and the potential miR-708 binding site on NRAS 3'UTR. Nucleotides mutated in miR-708 binding sites are shown in red. Right: data represent normalized means ± SD (n = 3 biological replicates). One-way ANOVA followed by Tukey's post-hoc test (B and C) and student's t-test (two-tailed) (D) were used for the statistical test (*p<0.05, **p<0.01, ***p<0.001, ****p<0.0001).

model [43]. Among all miRNAs with canonical binding sites on the *NRAS* 3'UTR (Figure S1 in S1 File), we chose miR-708 as our candidate to perform further molecular and functional analysis since miR-708 is a well-known tumor-suppressive miRNA in various cancer types [33]. To investigate whether miR-708 is a suitable drug for cancers driven by *NRAS* mutation, we collected three cell lines carrying *NRAS* mutation from different cancer types, SK-MEL-2, THP-1, and H1299, which belong to malignant melanoma, AML, and non-small cell lung cancer (NSCLC) respectively (Fig 1A).

To examine whether miR-708 can deplete NRAS, we measured NRAS protein levels in these three cell lines transfected with miR-708 (Figure S2 in S1 File). Significant downregulation of the NRAS protein level was observed in all cell lines with miR-708 overexpression through Western blot analysis (Fig 1B), as well as the mRNA level of *NRAS* in SK-MEL-2 cells (Fig 1C). A 3'UTR luciferase reporter assay was employed to confirm whether *NRAS* is directly regulated by miR-708. Wild type (WT) or mutated putative binding sequence of miR-708 on

the *NRAS* 3'UTR was inserted into a luciferase reporter construct (Fig 1D). Expression of miR-708 caused a significant decrease in luciferase activity in cells transfected with a luciferase reporter construct containing the WT *NRAS* 3'UTR, while this effect was rescued in cells expressing the mutant form (Fig 1D). These data suggest that *NRAS* is the direct downstream target of miR-708 and led us to determine how miR-708 modulates cellular activities through *NRAS* depletion in *NRAS* mutated cells.

## MicroRNA-708 exerts antitumor activities in NRAS-mutated cancer cells

To evaluate the function of miR-708 in different cancers carrying *NRAS* mutations, we transiently transfected SK-MEL-2, H1299, and THP-1 cells with miR-708 to evaluate the impact of miR-708 on uncontrolled cell proliferation driven by *NRAS* mutations. Expression of miR-708 impaired the proliferation rate of SK-MEL-2 and H1299 cells in the 2D clonogenic assay (Fig 2A, Figure S3A in S1 File). To assess the role of miR-708 in anchorage-independent cell growth, we seeded these three cell lines on soft agar and incubated those cells for 7–14 days. At the end of the experiment, we measured the number and area of colonies as indicators for survival and anchorage-independent cell division. The number and size of colonies in SK-MEL-2 cells transfected with miR-708 were significantly reduced compared to the control group (Fig 2B). Intriguingly, the results were different in THP-1 and H1299 cells, in which miR-708 overexpression only led to dramatic inhibition in the size of the colony but not in the colony number (Figure S3B, S3C in S1 File). To further explore the function of miR-708, we checked whether miR-708 modulates cell motility. We used Boyden chambers with or without Matrigel to assay invasiveness through the extracellular matrix and cell migration toward the chemoattractant concentration gradient. Three cell lines exhibited different metastatic behaviors when miR-708 was transiently overexpressed. We found that miR-708 effectively suppressed both the migration and invasion of SK-MEL-2 cells (Fig 2C). Similar to SK-MEL-2 cells, the motility of THP-1 cells was decreased with miR-708 overexpression (Figure S3D in S1 File). We only assayed the chemotaxis of THP-1 cells transfected with miR-708 because invasion is not frequently discussed in nonsolid cancer. On the other hand, H1299 behaved differently when miR-708 was transiently overexpressed. Specifically, invasion of H1299 cells was significantly inhibited by miR-708 while migration was only slightly inhibited in the presence of miR-708 (Figure S3E in S1 File).

Constitutively active *NRAS* is known to protect cells from apoptosis [44]; therefore, we also examined apoptosis in these three cell lines with miR-708 overexpression in the presence of the reactive oxygen species (ROS) donor, hydrogen peroxide ($H_2O_2$). Flow cytometry analysis of the early apoptosis marker Annexin V and late apoptosis marker 7-AAD showed that miR-708 transfection enhanced $H_2O_2$-triggered apoptosis (Annexin V positive) in SK-MEL-2 (Fig 2D) and H1299 (Figure S3F in S1 File) cells, but not in THP-1 cells (Figure S3F in S1 File). Taken together, these findings indicate that miR-708 exerts anticancer effects on *NRAS*-driven melanoma, AML, and NSCLC cell lines. To verify the relationship between *NRAS* depletion (Fig 1) and miR-708-mediated anticancer activities (Fig 2 and Figure S3 in S1 File), we next investigated whether cells with *NRAS* knockdown resulted in similar phenotypes as those presented in cells with miR-708 overexpression.

## MicroRNA-708 impairs cell proliferation and survival through the NRAS-mediated pathway in NRAS-mutated cancer cells

Melanoma is the major cancer type caused by *NRAS* mutation; therefore, we carried out *NRAS* depletion experiments in SK-MEL-2 cells. Due to the inhibition of cell proliferation, anchorage-independent growth, migration, and invasion by miR-708 in *NRAS* mutated cells (Fig 2

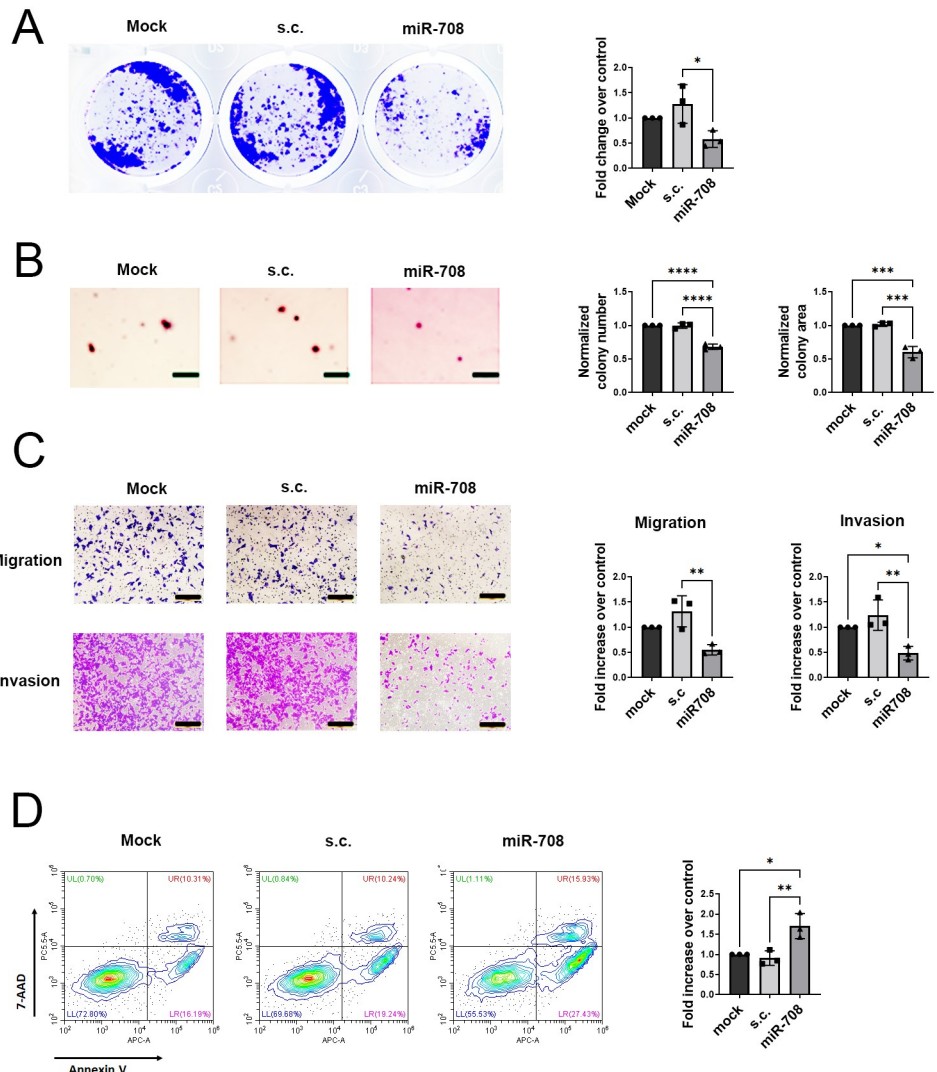

**Fig 2. MicroRNA-708 inhibits proliferation, anchorage-independent growth, motility, and cell survival in NRAS-mutated cancer cells.** (A) Clonogenic assay of SK-MEL-2 cells transfected with mock, s.c., and pre-miR-708. Left: representative images of SK-MEL-2 cells are shown. Right: histogram represents the fold change of mean ± SD (n = 3 biological replicates) relative to mock. (B) Soft agar colony formation assay was performed in SK-MEL-2 cells expressing mock, s.c., miR-708. Left: the representative images of SK-MEL-2. Right: the number and size of colonies were quantified and expressed as averaged fold increase compared to mock (n = 3 biological replicates). Scale bars: 200 μm (C) SK-MEL-2 cells expressing mock, s.c., miR-708 were subjected to transwell cell migration and invasion assay. Left: the representative images of migration and invasion. Scale bars: 200 μm. Right: the quantitative results of migration and invasion assay. Data are fold changes ± SD (n = 3 biological replicates) relative to mock. (D) Apoptosis of SK-MEL-2 cells overexpressing mock, s.c. and miR-708. Cells were incubated with 200 μM of $H_2O_2$ for 24 h and then examined by Annexin V/7-AAD staining through flow cytometry analysis. The apoptotic rate was represented by the percentage of Annexin V positive cells and then normalized with mock. Data are fold changes ± SD (n = 3 biological replicates) relative to mock. (A-D) One-way ANOVA followed by Tukey's post-hoc test was used for all statistical tests (*p<0.05, **p<0.01, ***p<0.001, ****p<0.0001).

and Figure S3 in S1 File), we are interested in finding out whether these miR-708-regulated cellular activities are through *NRAS* depletion. We used small interfering RNA (siRNA) to deplete *NRAS* expression and the silencing effect was validated by Western blot analysis (Fig 3A). SK-MEL-2 cells transiently transfected with siNRAS presented a lower proliferation rate and reduced anchorage-independent growth compared to control groups indicated by

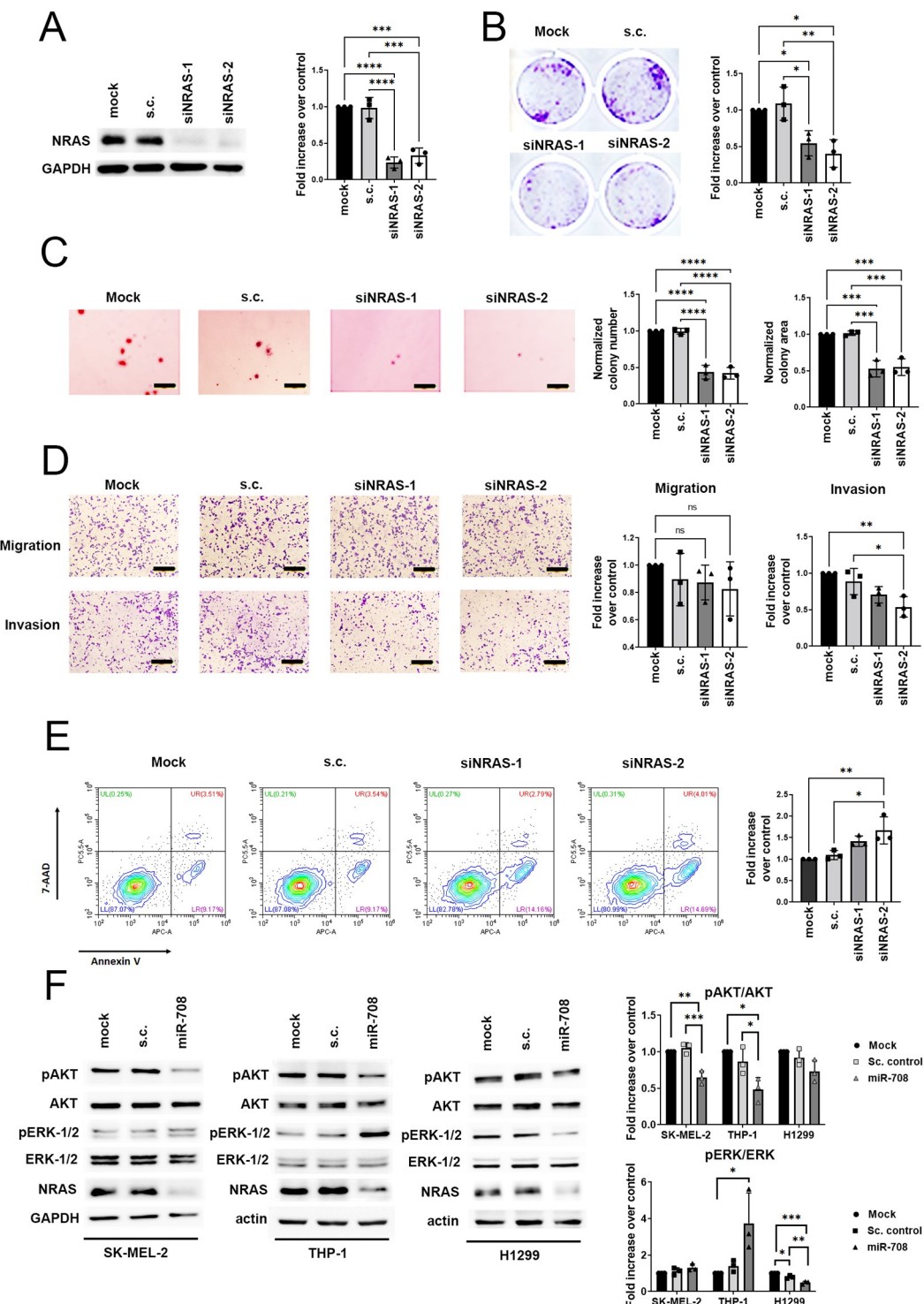

**Fig 3. MicroRNA-708 impairs cell proliferation through NRAS-mediated pathway.** (A) Left: Western blot analysis of SK-MEL-2 cells transfected with mock, s.c., siNRAS-1, and siNRAS-2. Right: quantitative analysis of NRAS protein levels, normalized with GAPDH level. Histogram show normalized mean ± SEM (n = 3 biological replicates) (B) Clonogenic assay of SK-MEL-2 cells transfected with mock, s.c., siNRAS-1, and siNRAS-2. Left: representative images of SK-MEL-2 cells are shown. Right: histogram represents the fold change of mean ± SD (n = 3 biological replicates) relative to mock. (C) Soft agar colony formation assay was performed in SK-MEL-2 cells expressing mock, s.c., siNRAS-1 and siNRAS-2. Left: the representative images of SK-MEL-2 cells in soft agar colony formation assay. Scale bars: 200 μm. Right: the number and size of colonies were quantified and expressed as averaged fold increase compared to mock ± SD (n = 3 biological replicates).

(D) SK-MEL-2 cells expressing mock, s.c., siNRAS-1 and siNRAS-2 were subjected to transwell cell migration and invasion assay. Left: the representative images of migration and invasion. Right: the quantitative results of migration and invasion assay. Scale bars: 200 μm. Data are the fold increase of means ± SD (n = 3 biological replicates) relative to mock. (E) Apoptosis of SK-MEL-2 cells overexpressing mock, s.c., siNRAS-1 and siNRAS-2. Cells were treated with 200 μM $H_2O_2$ 24 hours post-transfection and incubated for another 24 hours. The cells were then co-stained with Annexin V/7-AAD and subjected to flow cytometry analysis. The apoptotic rate was represented by the percentage of Annexin V positive cells and then normalized with mock. Data are fold changes ± SD (n = 3 biological replicates) relative to mock. (F) Left panel: Total cell lysate of SK-MEL-2, THP-1, and H1299 cells transfected with miR-708 and s.c. were collected and subjected to Western blotting to analyze the level of phosphorylation of AKT (pAKT) and ERK-1/2 (pERK-1/2), total AKT, total ERK-1/2, and NRAS. Right panel: The pAKT and pERK1/2 levels were quantified and normalized to total AKT and total Erk1/2 levels. Data presented as normalized mean ± SEM (n = 3 biological replicates). (A-F) One-way ANOVA followed by Tukey's post-hoc test was used for all statistical tests (*p<0.05, **p<0.01, ***p<0.001, ****p<0.0001).

clonogenic assay (Fig 3B) and soft agar colony formation assay (Fig 3C). Surprisingly, in contrast to miR-708 transfection, migration was not suppressed in cells with *NRAS* knockdown (Fig 3D). Meanwhile, invasion of SK-MEL-2 cells transfected with siNRAS was reduced to a lesser extent compared to miR-708 overexpression (Fig 3D). This result suggests that *NRAS* depletion by miR-708 may not play the central role in inhibiting cell migration and invasion. It is possibly dominated by other downstream targets mediated by miR-708, for example, Rap1B and NNAT, as reported previously in ovarian and breast cancer [35, 40]. Moreover, siNRAS transfection was also unable to fully recapitulate the effect of miR-708-enhanced ROS-induced apoptosis in SK-MEL-2 cells (Fig 3E), implying that miR-708 possibly induced cell apoptosis through additional targets, such as survivin [36]. To further elucidate the underlying mechanism of the miR-708-NRAS signaling pathway, we analyzed the activities of two dominant NRAS downstream effectors, the RAF-MEK-ERK cascade and the PI3K-AKT-mTOR pathway. Phosphorylation of AKT was significantly downregulated in miR-708-transfected SK-MEL-2 and THP-1 cells but modestly decreased in H1299 cells (Fig 3F). In contrast, phosphorylation of ERK protein in H1299 cells was significantly suppressed while that in SK-MEL-2 cells was barely affected. Unexpectedly, ERK phosphorylation was even 3-fold higher in THP-1 cells transfected with miR-708, showing three different responses of ERK activation in three different cell types (Fig 3F). Taken together, these findings demonstrate that miR-708 negatively regulates *NRAS* expression and subsequently suppresses one of the main effector pathways, to decelerate cancer cell proliferation, in the three cell lines we examined. On the other hand, it also indicates that the activation of these pathways in these cells may be not only dependent on NRAS activation but also rely on a much more complex mechanism.

## Cell proliferation is not affected by miR-708 in cells carrying wildtype NRAS

To further support the idea of miR-708 as a precision medicine for *NRAS* mutation-driven cancers, we overexpressed miR-708 in breast cancer cells and lung cancer cells carrying *KRAS* mutation, MDA-MB-231 and A549, as well as normal lung epithelial cells, BEAS-2B. In MDA-MB-231 and A549 cells, NRAS expression was greatly reduced (Fig 4A and 4C), whereas clonogenic cell proliferation was not suppressed by miR-708 overexpression (Fig 4B and 4D). As the proliferation of MDA-MB-231 and A549 relies on aberrant KRAS signaling, *NRAS* knockdown only results in minimal effects on the cells and thereby shows no growth defects. Similar to the data from MDA-MB-231 cells, cell proliferation was not significantly suppressed in BEAS-2B cells overexpressing miR-708 (Fig 4E and 4F). Overall, these results indicate that miR-708 exclusively exerts its tumor suppressive function on *NRAS* mutation-driven cancers. Collectively, our hypothesis that miRNA can act as a good candidate for *RAS* isoform-specific targeted therapy is feasible.

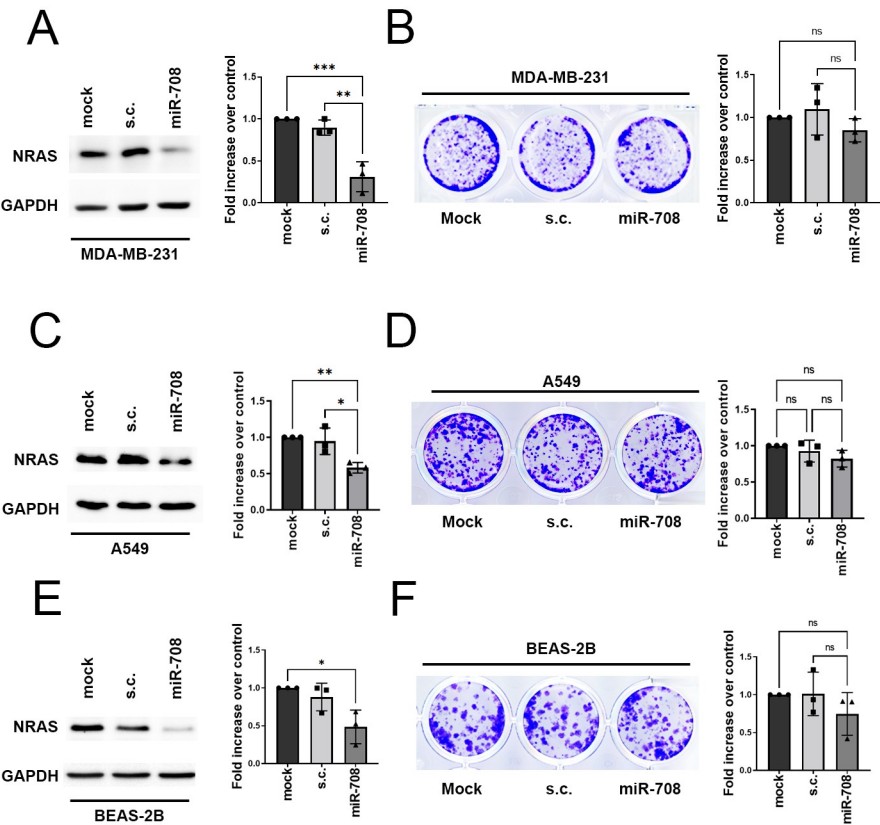

**Fig 4. MicroRNA-708 fails to inhibit proliferation in cells with wild-type NRAS.** (A) Left: Representative Western blot of NRAS protein expression in MDA-MB-231 cells transfected with mock, s.c., and pre-miR-708 for 48 h. Right: quantitative analysis of NRAS protein level, normalized with GAPDH. Histogram represents normalized means ± SEM (n = 3 biological replicates). (B) Clonogenic assay of MDA-MB-231 cells transfected with mock, s.c., and pre-miR-708. Left: representative images are shown. Right: histogram represents the fold change of mean ± SD (n = 3 biological replicates) relative to mock. (C) Left: Representative Western blot of NRAS protein expression in A549 cells transfected with mock, s.c., and pre-miR-708 for 48 h. Right: quantitative analysis of NRAS protein level, normalized with GAPDH. Histogram represents normalized means ± SEM (n = 3 biological replicates). (D) Clonogenic assay of A549 cells transfected with mock, s.c., and pre-miR-708. Left: representative images are shown. Right: histogram represents the fold change of mean ± SD (n = 3 biological replicates) relative to mock. (E) Left: Representative Western blot of NRAS protein expression in BEAS-2B cells transfected with mock, s.c., and pre-miR-708 for 48 h. Right: quantitative analysis of NRAS protein level, normalized with GAPDH. Histogram represents normalized means ± SEM (n = 3 biological replicates). (F) Clonogenic assay of BEAS-2B cells transfected with mock, s.c., and pre-miR-708. Left: representative images are shown. Right: histogram represents the fold change of mean ± SD (n = 3 biological replicates) relative to mock. (A-F) One-way ANOVA followed by Tukey's post-hoc test was used for the statistical test ($^*p<0.05$, $^{**}p<0.01$, $^{***}p<0.001$, ns: not significant).

## MicroRNA-708 presents moderate survival benefits in NRAS-mutated cancer patients

To investigate the prognostic value of miR-708 in *NRAS*-mutated cancer, we conducted a correlation analysis between miR-708 expression and patient survival with the TCGA database. Selected information from patients with *NRAS* single somatic mutation (#SSM) in the Skin Cutaneous Melanoma (SKCM), Acute Myeloid Leukemia (LAML), or Lung Adenocarcinoma (LUAD) dataset was applied for further analysis. Compared with the survival plot of patients among all SKCM, LAML, and LUAD, high expression of miR-708 presented a moderately better survival rate in patients with *NRAS* mutation from three different cancer types (Fig 5), especially in patients with LAML (Fig 5C) and LUAD (Fig 5E). Although the sample sizes for

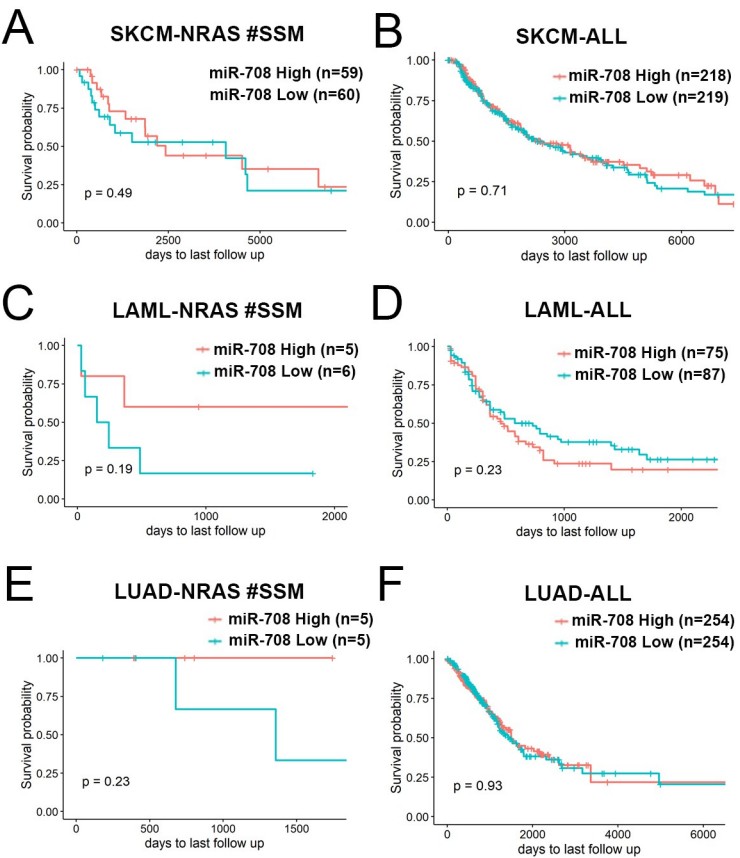

**Fig 5. MicroRNA-708 showed moderate survival benefits in NRAS-mutated cancer patients.** Retrospective analysis of Kaplan-Meier plots for miR-708 expression in association with overall survival. (A) 119 patients with Skin Cutaneous Melanoma (SKCM) carrying *NRAS* simple somatic mutation (#SSM). (B) 437 patients in all SKCM. (C) 11 patients from Acute Myeloid Leukemia (LAML) with *NRAS* #SSM. (D) 162 patients in all LAML. (E) 10 patients from Lung Adenocarcinoma with *NRAS* #SSM. (F) 508 patients in all LUAD. Patients were split into high and low expression groups based on the median expression of the miR-708. The log-rank test was used for the statistical analysis.

patients with *NRAS* mutation in LAML and LUAD are too small (n = 11 or 10, respectively) and that could lead to statistical bias, it is still encouraging to see the survival difference, supporting our hypothesis that miR-708 can serve as a therapeutic option in treating *NRAS* mutation-driven cancers.

## Discussion

In this work, miR-708 emerges as a novel therapeutic option for *NRAS* mutation-driven cancer which has long been considered a tough nut to crack in drug design. We showed that miR-708 directly suppressed *NRAS* expression and this *NRAS* depletion led to the reduced proliferation of cancer cells through subsequent oncogenic signaling inhibition. MicroRNA-708 restoration also weakened cancer cell motility in an *NRAS*-independent manner. Our findings provide convincing evidence for developing miR-708 as an *NRAS*-specific targeted therapy and we hope that patients who suffer from cancers driven by *NRAS* mutation will benefit from our research (Fig 6).

It is well documented that oncogenic NRAS signaling facilitates cells to degrade the surrounding extracellular matrix by enhancing its proteolytic activity and thus increases the

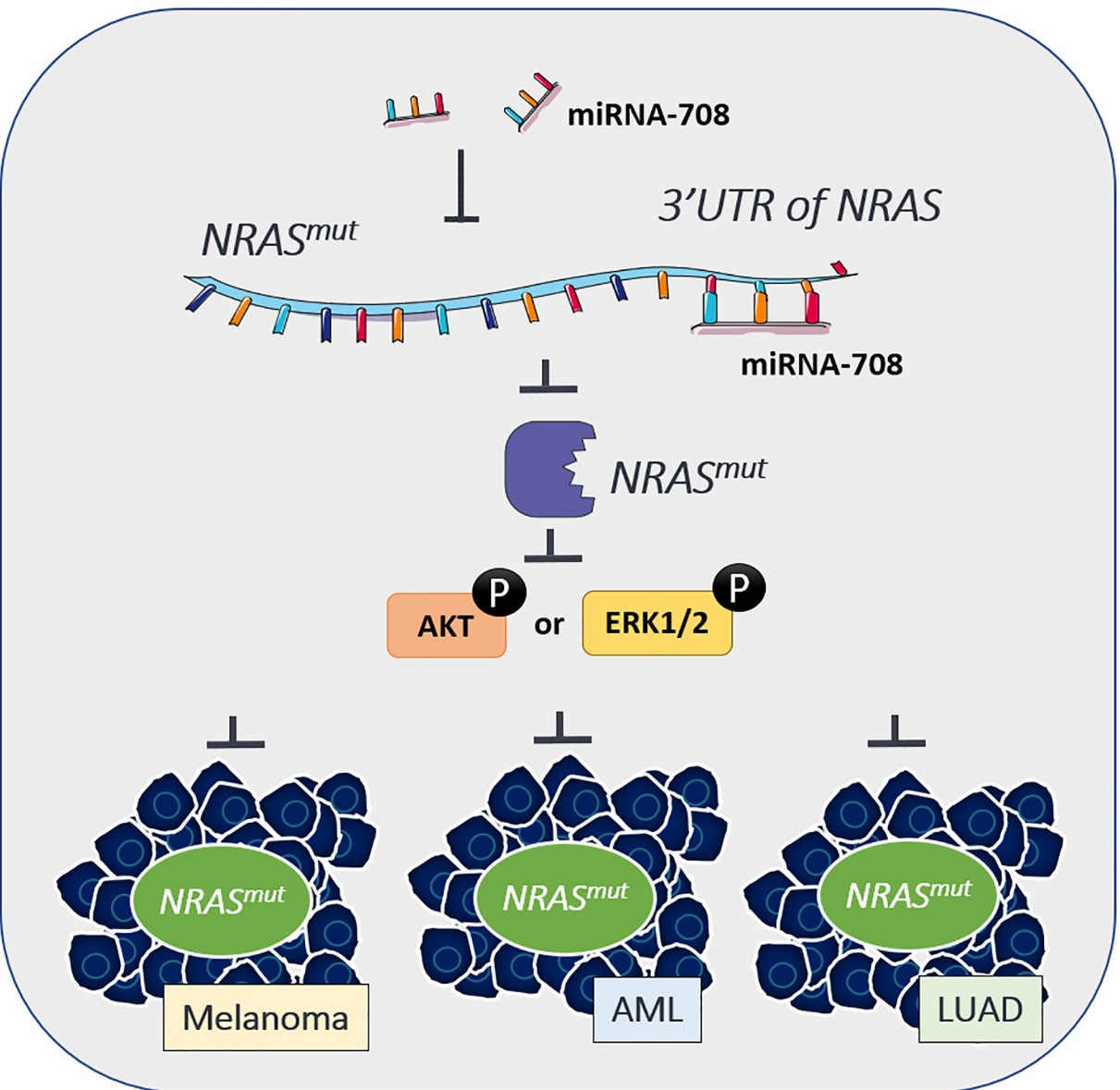

**Fig 6. Working model of miR-708 suppresses cancer progression in NRAS mutated cancers.** Expression of miR-708 depletes *NRAS* and impairs downstream PI3K/AKT/mTOR or RAF/MEK/ERK activation, resulting in the suppression of cell proliferation, anchorage-independent growth, and enhanced reactive oxygen species-induced apoptosis in *NRAS*-mutation driven cancer cells. MicroRNA-708 possesses the potential to be developed as the targeted therapy for *NRAS* mutant melanoma, AML, LUAD et cetera.

invasiveness of *NRAS*-mutated melanoma cells [45]. Our results echo those of previous studies in which *NRAS* knockdown moderately suppressed the invasion of melanoma cells, while migration was not affected (Fig 3D). By taking advantage of miR-708 as a powerful tumor suppressor by targeting numerous cancer-initiating or cancer-facilitating genes [33], we could use miR-708 to combat cancers from multiple aspects in addition to NRAS inhibition, resulting in a net positive outcome. This benefit is confirmed by strongly abrogated migration and invasion in miR-708-transfected cells (Fig 2C and Figure S3D, S3E in S1 File) and promotion of ROS-induced apoptosis (Fig 2D, Figure S3F in S1 File), instead of only mildly reduced invasiveness and mildly induced apoptosis in cells with *NRAS* knockdown (Fig 3D and 3E). Taken

together, the fact that we found miR-708 can effectively reduce the metastatic potential of *NRAS*-mutated cancer suggests a potential therapeutic strategy to treat *NRAS* mutation-driven tumorigenesis.

In this study, we have tried to uncover the underlying signaling transduction regulated by the miR-708/NRAS axis. Given that ERK and AKT are both downstream effectors of RAS, our results were intriguing as their activation was differentially regulated when NRAS was suppressed by miR-708 in different cancer cell lines. In particular, we observed that AKT activation was significantly inhibited in miR-708-transfected SK-MEL-2 and THP-1 cells while ERK activation was dominantly suppressed in H1299 cells overexpressing miR-708 (Fig 3F). The differential downstream effector responses of NRAS to miRNA-708 overexpression in different cell lines could be due to the following factors. First, miR-708 is a microRNA that can modulate differential gene expression by targeting the 3' UTR of specific mRNAs to stimulate translational inhibition or mRNA degradation. The regulatory mechanism of miR-708 in different cells may be incomparable since different cells have alternative 3'UTR isoforms, which may affect the inclusion of microRNA binding sites and subsequently affect the gene repression efficiency [46]. Second, the effector concentration may be the key factor that affects NRAS binding and presumably leads to different signaling networks in different cell types [47]. Third, the expression levels of miR-708 target mRNAs can vary between different cell lines, leading to the different regulation of NRAS downstream effectors, such as epigenetic modification, transcriptional regulation, or posttranslational modifications. Overall, the differential downstream effector responses of NRAS in different cell lines are likely due to the combination of all the abovementioned reasons. Further investigations such as proteomic analysis are needed to provide a better idea of miR-708-NRAS regulation in different cell lines. By inference, we speculated that the compensatory mechanism was triggered to sustain growth, proliferation, and survival signals in *NRAS*-driven malignancies when *NRAS* was blocked. Although the exact regulation of miR-708 on the NRAS signaling pathway remains inconclusive, we believe that there is an opportunity for miR-708 to cooperate with ERKi and AKTi to tackle *NRAS* mutant cancer effectively.

Previous studies on *NRAS* knockout mice showed that *NRAS* is dispensable during normal development [48], revealing that there might be some functional redundancy between *NRAS* and *HRAS* isoforms [49]. Moreover, when nanoparticle-encapsulated miR-708 was distributed in several major organs, no toxicity was observed, indicating that elevated miR-708 levels have no impact on normal tissues [50]. Here our data further confirmed that depletion of NRAS protein level in normal lung epithelial cells did not disturb cell proliferation (Fig 4E and 4F), echoing the previous study showing miR-708 has no impact on the proliferation of normal lung cell line, WI-38 [51]. Overall, it is suggested that using miR-708 to suppress *NRAS* mutation-driven cancer will be a safe and less toxic approach for standard *NRAS*-targeted therapy.

Several microRNAs have been shown to act as tumor suppressors by targeting *NRAS* in various cancer types, including breast [52], lung [53], colorectal [54], and prostate cancers [55]. However, all these works were restricted to a single type of cancer. Moreover, the current concept of precision cancer treatment is based on the genetic profile of a cancer patient and provides the most accurate gene-specific but not organ-specific treatment [56]. Therefore, we tested our hypothesis of using miR-708 to inhibit *NRAS* mutation-driven cancer in three types of cancer cell lines: melanoma, AML, and NSCLC. Our study presents a proof of concept in using one microRNA to combat multiple cancers harboring the same mutation. In our case, it was a *NRAS* mutation.

In reality, precision medicine often involves analyzing smaller subsets of patients with specific genetic mutations or other unique characteristics to identify targeted treatments. In cases where the incidence of a particular mutation is low, it may be difficult to obtain large sample

sizes for analysis. As in our case, we analyzed patients from the TCGA database with *NRAS* mutations, including SKCM (n = 437), LUAD (n = 508), and LAML (n = 162). The rates of *NRAS* mutation in these cancer types are 20%, 0.9%, and 15%, respectively. Therefore, we can only collect a very limited number of patients with *NRAS* mutations from each disease (Fig 5), and the analyzed result may lead to sampling bias. However, we still believe that analyzing all available patients with *NRAS* mutations in the TCGA database is valuable in revealing the potential for using microRNA to treat *NRAS*-mutation-driven cancers. Overall, while small sample sizes can present challenges in precision medicine research, they should not prevent us from exploring potential treatment options for specific patient populations.

The success of miR-708 suppressing *NRAS* is proof of the concept that using microRNA to antagonize the "undruggable" cancer target. Similar to *NRAS*, many oncogenes in cancer are still deemed intractable. These targets are difficult to drug with conventional pharmacological inhibitors because of the lack of deep pockets [1], large protein-protein interactions [55], or the druggable coding domain is spliced out [57]. Some notorious examples of undruggable proteins include KRAS, MYC, and androgen receptor variant 7 (AR-V7) [58]. Therefore, applying our strategy extensively to all these "yet to be drugged" targets by using microRNA to deplete their mRNA transcripts or block their translation is an alternative option to target cancers driven by these undruggable oncogene mutations.

## Conclusion

In summary, our study provides new insights regarding the use of microRNA to overcome the limitation of small molecule- or protein-based drugs to target those traditionally inaccessible mutated proteins. By demonstrating that miR-708 can effectively suppress *NRAS*-mutated cancer, including melanoma, AML, and NSCLC progression, we offer an opportunity to use miR-708 as precision medicine for *NRAS* mutation-driven cancer, and further explore a new era of miRNA-based drug development in precision cancer therapy.

## Supporting information

**S1 File.**
(PDF)

**S2 File. Clinical information of SKCM, LUAD, and LAML patients and miR-708 expression data retrieved from the TCGA cohort.**
(XLSX)

## Author Contributions

**Conceptualization:** Jia Meng Pang, Kai-Ti Lin.

**Data curation:** Po-Chen Chien, Ming-Chien Kao, Pin-Xu Chen, Yu-Ling Hsu.

**Funding acquisition:** Kai-Ti Lin.

**Investigation:** Jia Meng Pang, Po-Chen Chien, Ming-Chien Kao, Pin-Xu Chen, Yu-Ling Hsu, Chengyang Liu, Xiaowei Liang, Kai-Ti Lin.

**Methodology:** Jia Meng Pang, Kai-Ti Lin.

**Software:** Pei-Yun Chiu.

**Supervision:** Kai-Ti Lin.

**Validation:** Jia Meng Pang, Kai-Ti Lin.

**Visualization:** Jia Meng Pang.

**Writing – original draft:** Jia Meng Pang.

**Writing – review & editing:** Kai-Ti Lin.

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
