## [Decision Letter · Decision Letter 0]

10 Feb 2023

PONE-D-23-00076MicroRNA-708 emerges as a potential candidate to target undruggable NRASPLOS ONE

Dear Dr. LIN,

Thank you for submitting your manuscript to PLOS ONE. After careful consideration, we feel that it has merit but does not fully meet PLOS ONE’s publication criteria as it currently stands. Your manuscript has been read by two well qualified reviewers. As you can see from the reviewers' comments below, the reviewers raised significant concerns on the quality of the manuscript. The manuscript would require revisions and new experiments to strengthen the conclusions. In addition, the reviewers also showed concerns on the statistics and English problems. Therefore, we invite you to submit a revised version of the manuscript that addresses the points raised during the review process.

We look forward to receiving your revised manuscript.

Kind regards,

Ming Tan

Academic Editor

PLOS ONE

Journal Requirements:

"We are grateful to National Tsing Hua University for funding support our Competitive Research Team (Project No. 110Q2729E1 and 111Q2713E1) through the Ministry of Education's Higher Education Sprout Project."

Reviewers' comments:

Reviewer's Responses to Questions

**Comments to the Author**

1. Is the manuscript technically sound, and do the data support the conclusions?

Reviewer #1: Partly

Reviewer #2: Partly

2. Has the statistical analysis been performed appropriately and rigorously? 

Reviewer #1: Yes

Reviewer #2: No

3. Have the authors made all data underlying the findings in their manuscript fully available?

Reviewer #1: Yes

Reviewer #2: No

4. Is the manuscript presented in an intelligible fashion and written in standard English?

Reviewer #1: No

Reviewer #2: Yes

5. Review Comments to the Author

Reviewer #1: The present manuscript describes a report using miRNA-708 that targets the NRAS gene to develop a miRNA-based management of NRAS mutation-driven cancers. They show that NRAS is a direct target of miRNA-708 which reduce NRAS protein levels in cell lines with NRAS mutation (melanoma, leukemia, and lung cancer), resulting in suppressed cell proliferation, anchorage independent growth, and promotion of reactive oxygen species-induced apoptosis. Its results also shown that the activities of NRAS-downstream effectors, PI3KAKT-mTOR or RAF-MEK-ERK signaling pathway, were affected in miR-708 over-expressing cells. Moreover, they shown that cell proliferation was not disturbed by miRNA-708 in cell lines carrying wild-type NRAS. They conclude that their results unveil the therapeutic potential of miRNA-708 in NRAS mutation-driven cancers as a precision medicine in cancer treatment.

In my opinion the study is well conducted and the results support the claimed conclusions. However, I believe that the manuscript can be improved resolving the following concerns.

Made clear in the whole manuscript when the symbols are referred to proteins or its genes…

Write properly the symbols of genes described (italic). Ej. RAS, NRAS and HRAS in the whole document when refers to gene names…

The last sentence in Survival Correlation Analysis in Material & Methods section is incomplete (page 13, line 249).

Fig 5 A, properly indicate the legend…

Respect to the results of the miRNA-708 over-expression on the activities of NRAS-downstream effectors, PI3KAKT-mTOR or RAF-MEK-ERK signaling pathway, a paragraph with information about other miRNA-708 targets that could explain the results on the activity of the ERK in SK-MEL-2 and TPH-1 cell lines should be included.

The survival-miRNA-708 expression correlation study results should be included in the discussion section…

I recommend the review of the manuscript by a native English language speaker

Reviewer #2: 1. Based on the experimental evidence provided by the authors, it is not clear whether NRAS is a direct or indirect target of mir-708. Is there any experimental evidence of NRAS and mir-708 interaction?

2. The authors should have included another control cell line with normal NRAS expression in order to draw the conclusion that the overexpression of miRNA-708 decreased the NRAS protein levels in the cell lines with NRAS mutation.

3. Overexpression of mir-708 only has a considerable anti-tumour impact on SK-ME-2 cells compared to the other two cell lines. Not all of the metrics taken into consideration for THP-1 and H1299 cells show a significant anti-tumour impact. In the result section (lines 308-310), the assertion drawn from the experiments does not indicate the same.

4. The lack of a robust anti-tumour effect from NRAS knockdown in SK-ME-2 raises the question of whether it is a potent therapeutic target.

5. What could be the plausible explanation for the differential response of mir-708 overexpression towards downstream effectors of NRAS?

6. In the Kaplan-Meier plot, the P value is quite high for the data to be considered significant. Thereby the assertions drawn may not hold true.

6. PLOS authors have the option to publish the peer review history of their article (what does this mean?). If published, this will include your full peer review and any attached files.

Reviewer #1: **Yes: **Homero Reyes de la Cruz

Reviewer #2: No

---

## [Author Response · Author response to Decision Letter 0]

27 Mar 2023

The file of respond to reviewers is attached in the submission profiles.

---

## [Decision Letter · Decision Letter 1]

10 Apr 2023

MicroRNA-708 emerges as a potential candidate to target undruggable NRAS

PONE-D-23-00076R1

Dear Dr. LIN,

We’re pleased to inform you that your manuscript has been judged scientifically suitable for publication and will be formally accepted for publication once it meets all outstanding technical requirements.

Kind regards,

Ming Tan

Academic Editor

PLOS ONE

Additional Editor Comments (optional):

The manuscript is acceptable now. One of the reviewers mentioned a few minor editorial issues. I think these can be corrected during copy editing or proofreading.

Reviewers' comments:

Reviewer's Responses to Questions

**Comments to the Author**

1. If the authors have adequately addressed your comments raised in a previous round of review and you feel that this manuscript is now acceptable for publication, you may indicate that here to bypass the “Comments to the Author” section, enter your conflict of interest statement in the “Confidential to Editor” section, and submit your "Accept" recommendation.

Reviewer #1: All comments have been addressed

2. Is the manuscript technically sound, and do the data support the conclusions?

Reviewer #1: Yes

3. Has the statistical analysis been performed appropriately and rigorously? 

Reviewer #1: Yes

4. Have the authors made all data underlying the findings in their manuscript fully available?

Reviewer #1: Yes

5. Is the manuscript presented in an intelligible fashion and written in standard English?

Reviewer #1: Yes

6. Review Comments to the Author

Reviewer #1: I believe that the authors took into account all the observations made to the manuscript and from my point of view they have made the appropriate corrections and added the suggestions, so I believe that the manuscript has improved considerably and can now be accepted for publication. However, the manuscript still has minimal editing errors that can be corrected. E.j., Page 4, line 65; “RAS” should be in normal (not cursive) letters; a space is missing, line 180; write properly 4oC, line 197…

7. PLOS authors have the option to publish the peer review history of their article (what does this mean?). If published, this will include your full peer review and any attached files.

Reviewer #1: **Yes: **Reyes de la Cruz, Homero

---

## [Editor Report · Acceptance letter]

13 Apr 2023

PONE-D-23-00076R1 

MicroRNA-708 emerges as a potential candidate to target undruggable NRAS 

Dear Dr. Lin:

I'm pleased to inform you that your manuscript has been deemed suitable for publication in PLOS ONE. Congratulations! Your manuscript is now with our production department. 

Kind regards, 

on behalf of

Dr. Ming Tan 

Academic Editor

PLOS ONE